

# Strong modification of stratospheric ozone forcing by cloud and sea ice adjustments

Y. Xia[1], Y. Hu[2], and Y. Huang[1]

[1]Department of Atmospheric and Oceanic Sciences, McGill University, Montreal, Canada
[2]Department of Atmospheric and Oceanic Sciences, Peking University, Beijing, China

*Correspondence to*: Y. Xia (yan.xia3@mail.mcgill.ca)

**Abstract.** We investigate the climatic impact of stratospheric ozone recovery (SOR) with a focus on the surface temperature change in atmosphere-slab-ocean coupled climate simulations. We find that although SOR would cause significant surface warming (global mean: 0.2 K) in a climate free of clouds and sea-ice, it may result in surface cooling (-0.06 K) in the real

climate. The results here are especially interesting in that the stratosphere-adjusted radiative forcing is positive in both cases. Radiation diagnosis shows that the surface cooling is mainly due to a strong radiative effect resulting from significant reduction of global high clouds and, to a lesser extent, from an increase in high-latitude sea ice. Our simulation experiments suggest clouds and sea ice are sensitive to stratospheric ozone perturbation, which constitutes a significant radiative adjustment that influences the sign and magnitude of the global surface temperature change.

**1 Introduction**

Observational records show that stratospheric ozone has declined prior to the late 1990s and then started stabilizing and even slow increasing, especially in the Polar Regions (WMO, 2007, 2011). It is expected that the ozone layer would return to the pre-1980 level in the 2050s (Bekki, 2011). It is known that ozone is a greenhouse gas, and that stratospheric ozone has a warming effect on tropospheric-surface climate, which has been demonstrated by early simulation works with radiative-

convective models (Ramanathan and Dickinson, 1979;Lacis et al., 1990). Consistent with such understanding, ozone depletion generally leads to a negative radiative forcing (after accounting for stratospheric temperature adjustment) that cools the climate (Forster and Shine, 1997;Hansen et al., 2005;Conley et al., 2013;Myhre et al., 2013;Macintosh et al., 2016). On such basis,



one would expect that stratospheric ozone recovery (SOR) exerts a positive forcing that should lead to troposphere and surface warming. The single-column simulation by Hu et al. (2011) agrees with such expectation, although their efforts to distinguish the responses to SOR in full general circulation models (GCMs) is impeded by climate sensitivity differences between the two groups of models (McLandress et al., 2012). Very interestingly, McLandress et al. (2012) show a weak troposphere-surface cooling in response to SOR in a coupled chemistry-climate model (CCM). As presented below, such a weak cooling is also seen in our simulation with an atmospheric GCM coupled to a slab-ocean model. These results raise important questions: how does surface cooling result from the positive radiative forcing of SOR in GCM simulations? Why do GCMs and radiative-convection models yield opposite results? In this paper, we are motivated to answer these questions and reconcile the contradiction of the warming prediction based on single-column model simulations.

One prominent deficiency of the one-dimensional radiative-convective models is that they neglect effects of clouds as well as snow and ice albedo. Thus, results from these simplified models may not realistically represent the responses to SOR. Hence, our hypothesis is that the radiative adjustment of clouds and sea ice may override the forcing of SOR and change the direction of surface temperature change in more sophisticate GCMs. To test this hypothesis, we perform two sets of SOR forcing experiments using a three-dimensional climate model, one with standard settings and the other with cloud and sea-ice artificially removed in the simulation. Comparison of the two sets of simulations shall elucidate the effects of cloud and sea ice. In the following sections, we will describe the configuration and results of these experiments, dissect the simulations from a radiative budget perspective, and summarize our main findings in order.

## 2 Model and experiment design

Here, we conduct and analyze a series of SOR experiments using the NCAR Community Atmosphere Model, version 3 (CAM3) coupled with a Slab Ocean Model (SOM) (Collins et al., 2006;Neale et al., 2010). All of the runs presented below are made with T42 horizontal resolution (~2.8°x2.8°) and coupled to a 50-meter-deep SOM. The SOM configuration uses a simple ocean component (Kiehl et al., 2006;Danabasoglu and Gent, 2009), combined with a thermodynamic sea ice component that is based on the Community Sea Ice Model (CSIM5, (Briegleb, 2004)) and allows for a fully-interactive treatment of surface exchange

processes in CAM3. Danabasoglu and Gent (2009) compare the slab ocean and the fully coupled configurations of CCSM3 and find that the slab ocean setup provides a good estimate of the climate sensitivity of the fully coupled model. Although the slab-ocean component lacks explicit representation of ocean currents, GCM surface winds drive the sea ice dynamics, with advection simulated as a cavitating fluid (Flato and Hibler, 1990, 1992). Compared with the coupled atmosphere-ocean

simulations by CESM1 (CAM5), the annual cycle of climatological sea ice extent has similar magnitude (varying from 3 to $15 \times 10^6$ km$^2$) in SOM. The variabilities of the annual-mean sea ice extent are also similar (about 2-3$\times 10^6$ km$^2$) in SOM and coupled atmosphere-ocean simulations.

In order to isolate the effect of clouds and sea ice, two sets of experiments are conducted here. In the first set, we use standard settings of the model, without any modification of cloud and sea ice. In the other set of integrations, we set the freezing

temperature to -180 degree centigrade so that there is effectively no sea ice in the simulation. We also set all the cloud fractions to zero in radiative heating rate and flux calculations and thus suppress the radiative effects of clouds. To restore radiative energy balance, following Koll and Abbot (2013) we reduce the solar constant by 120 W m$^{-2}$, because CAM3 has a global mean cloud forcing of ~30 W m$^{-2}$. These two sets of experiments are denoted as "Standard" and "No Cloud No Sea-Ice (NCNSI)" respectively in the following. The global and climatological mean surface temperature is 291.4 K in the NCNSI

experiment, which is comparable to the climatology in the Standard experiment (about 2 K warmer). Note that the cloud modification used here does not affect the generation of clouds in GCM integration or related latent heating of the atmosphere. The hydrological cycle, as reflected by the climatology of precipitation, in the NCNSI experiment is similar to that in the Standard experiment. Thus, the NCNSI simulation provides a reasonable hypothetical world for comparing the radiative responses to SOR.

In order to examine the impact of SOR on surface temperature, two 100-year integrations, prescribed with identical concentrations of well-mixed greenhouse gases ($CO_2$, $CH_4$, $N_2O$, etc.) but different stratospheric ozone concentrations, are conducted in both the standard and the NCNSI experiments. The monthly mean ozone volume mixing ratios averaged over 1999-2003 (scenario 2000) and 1979-1983 (scenario 1979), taken from the ERA-Interim reanalysis (Dee et al., 2011), are

prescribed in these two integrations to represent "present" (2000) and "recovery" (1979) scenarios respectively. In order to eliminate the influence of tropospheric ozone, the ozone below 200 hPa in the recovery scenario is fixed at scenario 2000 level. To see the impact of SOR, the idealized ozone change in the recovery scenario above 200 hPa is set to the absolute value of the difference between 1979 and 2000 (Figure 1 a). In comparison, the ozone in the recovery scenario increases by about 28 Dobson Unit (DU) in the tropics and subtropical regions, about 63 DU in Arctic and about 73 DU in Antarctic (Figure 1 b). Both scenario experiments are initialized from an equilibrated present-day CAM3 simulation with Sea surface temperature (SST) prescribed to be the climatological mean values of the period 1980-2000. The atmospheric states in all these experiments approach steady states after 10 years of integration. We assess the SOR impacts by contrasting the means of appropriate variables in the last 90 years of two 100-year simulations (the difference between two equilibrium states).

## 3 Surface temperature change

As shown by Figure 2, SOR causes noticeable changes in not only stratospheric but also tropospheric and surface climate. The stratosphere in both the standard and NCNSI experiments is significantly warmed, as expected from the radiative heating effect of stratospheric ozone. On the other hand, SOR leads to tropospheric and surface warming in the NCNSI experiment, while noticeable cooling is seen in the Standard experiment (compare Figure 2 a and b). The global and annual mean surface temperature change is +0.2 K and -0.06 K in the two experiments, respectively. The surface warming in the NCNSI experiment occurs in all seasons and at most latitudes. In comparison, surface cooling in the Standard experiment is the strongest in the two polar regions (reaching -0.8 K in Arctic in boreal autumn), and is also strong (about -0.2 K) over the high-latitude Southern Oceans (40ºS-70ºS).

The results here support our hypothesis that the different responses to SOR (cooling vs. warming) are caused by clouds and sea ice. It is interesting that the same SOR perturbation drives surface climate changes in opposite directions due to effects of clouds and sea ice. This is especially interesting because the stratosphere-adjusted forcing of SOR (as detailed in the following section) is similar (positive) in the NCNSI and Standard experiments.

## 4 Radiation diagnosis



### 4.1 Instantaneous forcing

We calculate the instantaneous radiative forcing (IRF) of SOR using a radiative transfer model, RRTMG (Mlawer et al., 1997). The radiative forcing is calculated as the change in top-of-atmosphere (TOA) radiation fluxes in response to the stratospheric ozone change (from 2000 to 1979 values) at every grid box using monthly-mean temperature, water vapor, and cloud profiles from a 2000 equilibrium integration. Following Cronin (2014), we use the insolation-weighted method to calculate the monthly-mean solar zenith angle. The global and annual mean forcing values are provided in Table 1. Due ozone absorption of shortwave solar radiation (mainly in the 200-315 nm UV region) and longwave terrestrial thermal emission (mainly around 9.6 μm), the SOR as prescribed in our experiments induces a *positive* (downward at TOA, i.e., warming) forcing in both the NCNSI and Standard experiments. The global mean values are 0.49 W m$^{-2}$ and 0.60 W m$^{-2}$, respectively. In both experiments, the forcing has a flat zonal mean pattern, due to compensating effects of the latitudinal variations in surface thermal radiation and ozone concentration. In contrast, the shortwave SOR forcing peaks at two poles, which is caused by the higher local ozone concentration.

### 4.2 Stratospheric adjustment

Ozone heats the stratosphere due to its absorption of solar radiation. Here, the stratospheric adjustment, i.e., the radiative impact due to stratospheric warming in response to SOR, is calculated using a kernel method, following Zhang and Huang (2014) and Huang et al. (2015). The stratospheric temperature kernels of Shell et al. (2008) are used here. The stratospheric temperature change is calculated as the temperature difference between the 1979 and 2000 equilibrium integrations. As higher stratospheric temperatures mean more thermal radiation radiated to the space, stratospheric adjustments evaluated here are negative in both experiments. Nevertheless, the stratosphere-adjusted forcing (SAF, i.e., instantaneous forcing plus stratospheric adjustment) remains positive in both NCNSI (0.30 W m$^{-2}$) and Standard experiments (0.29 W m$^{-2}$). In addition, we also calculate the SAF with RRTMG using the fixed dynamical heating method (Ramanathan and Dickinson, 1979), and find the SAF in the Standard experiments to be 0.21 W m$^{-2}$, which is in agreement with the kernel method. Note that as discussed by Huang et al. (2015), the adjusted forcing evaluated using TOA flux equals that evaluated using tropopause flux



if the stratosphere adjusts to a radiative equilibrium. The fact that the stratosphere-adjusted forcing is positive indicates that the weak cooling in the Standard experiment is not predictable from SAF, but is influenced by tropospheric adjustments.

### 4.3 Tropospheric adjustments

Here we analyze the radiative contributions by other atmospheric and surface variables, namely temperature, water vapor, sea-ice (albedo) and clouds, mainly using the kernels of Shell et al., (2008). Note that the radiative effect of clouds is obtained using the cloud forcing adjustment method that incorporates the instantaneous forcing and stratospheric adjustment calculated above (c.f. Zhang and Huang (2013)).

In the Standard experiment, we find the radiative effects of clouds and sea-ice to be strongly negative ($-0.39$ and $-0.10$ W m$^{-2}$, respectively; see Table 1). The cloud effect consists of $-0.26$ W m$^{-2}$ in the longwave and $-0.13$ W m$^{-2}$ in the shortwave. This offsets the warming effect of SOR forcing (a SAF of 0.29 W m$^{-2}$). As a result, there is a weak global cooling in surface temperature ($-0.06$ K). The radiation budget is reinstalled by the positive radiation changes (reduction of outgoing radiation) caused by the surface cooling (0.08 W m$^{-2}$) and by atmospheric temperature and water vapor changes ($-0.04$ and 0.10 W m$^{-2}$, respectively).

In order to separate the fast adjustments in the troposphere from surface temperature-related feedback effects, we conduct a SOR experiment using CAM3 with fixed SST and sea ice (Fixed-SST/SI). Two simulations forced with prescribed climatological SST and SI averaged over the years 1980-2000 are performed with different ozone concentrations as described above. The stratosphere and troposphere-adjusted forcing (effective radiative forcing, ERF) is obtained by contrasting the averages over the last 15 years of the two 35-year integrations. The ERF is found to be 0.01 W m$^{-2}$, consisting of an instantaneous forcing of 0.60 W m$^{-2}$, a stratospheric adjustment of $-0.31$ W m$^{-2}$, and a tropospheric adjustment of $-0.28$ W m$^{-2}$ (which is mainly contributed by clouds: $-0.25$ W m$^{-2}$) (Table 1). Evident from these results, the cloud radiative effect in the Standard experiment is largely a tropospheric adjustment, which together with the stratospheric adjustment offsets instantaneous forcing of ozone and results in a neutralized ERF.





In comparison, in the NCNSI experiment, without the offsetting negative radiative effects of clouds and sea ice, a significant global warming (0.2 K) results from the SOR forcing, which gives rise to a radiative effect of -0.26 W m$^{-2}$. The water vapor feedback in this experiment is strong and positive (0.77 W m$^{-2}$), although it is offset by the atmospheric temperature feedback (-0.77 W m$^{-2}$).

In summary, these results show that significant radiative cooling effects caused by the adjustments of clouds and sea ice in response to SOR explains the weak global cooling in the Standard experiment.

### 4.4 Surface radiation budget

Complementary to the TOA radiation budget decomposition, we also analyze the surface radiation flux change driven by SOR. Figure 4 shows the changes in the surface radiation budget from the 2000 equilibrium integration relative to the 1979

equilibrium integration. The changes in the net surface shortwave radiation in both experiments can be explained by ozone absorption of UV radiation. In the NCNSI experiment, the global and annual mean reduction is -0.60 W m$^{-2}$. The maximum reduction reaches -2.4 W m$^{-2}$ in the Northern Hemisphere and -1.6 W m$^{-2}$ in the Southern Hemisphere. Both occur at high latitudes in summer because of the largest stratospheric ozone increases there. In the Standard experiment, the global and annual mean reduction is -0.62 W m$^{-2}$. Compared to the NCNSI experiment, the duration and spatial coverage of the net

shortwave radiation change is also significantly modified by clouds and sea ice (Figure 4 g and j). Here we measure the cloud radiative effect (CRE) by the difference between the all-sky and clear-sky surface radiation. The changes in longwave and shortwave CRE in response to SOR are shown separately, with global and annual mean values of -0.26 W m$^{-2}$ and 0.04 W m$^{-2}$, respectively. The radiative effect of sea ice is measured as the surface radiation change caused by surface albedo change, i.e., climatologic surface downward shortwave radiation times surface albedo change. The global mean shortwave radiation

change due to albedo change is measured to be -0.11 W m$^{-2}$.

The greenhouse effect of ozone enhances the surface downward longwave radiation. This enhancement is augmented by the atmospheric warming and moistening in the NCNSI experiment, which altogether overrides the cooling effect of ozone in the shortwave (Figure 4 c). The global and annual mean net radiation change is +1.1 W m$^{-2}$. This explains the surface warming in



this experiment. In comparison, the enhancement in the downward longwave radiation in the Standard experiment is less strong and limited to low latitude regions. This is mainly because of a strong negative change in cloud forcing (Figure 4 h). The global and annual mean net radiation change is -0.72 W m$^{-2}$, which explains the global cooling in this experiment.

In summary, the surface temperature responses in both experiments (Figure 2 a and b) are consistent with the changes in the net radiation at the surface (Figure 4 c and f). The comparison between the NCNSI experiment and the Standard experiment again highlights impacts of clouds and sea ice on the radiation budget, which can override the initial radiative perturbation of ozone and lead to different surface temperature responses. We will elaborate this point in the following section.

**5 The roles of cloud and sea ice**

Figure 2 e shows the response of the cloud fraction in the Standard experiment. There is general reduction in cloud fraction, especially for those high clouds near the tropopause. The decrease in high clouds is associated with a decrease in relative humidity caused by the SOR warming of the upper-troposphere and lower-stratosphere (Jenkins, 1999;Yang et al., 2012). This then accounts for the aforementioned negative TOA longwave cloud radiative effect (Table 1; Figure 3) and the negative change in CRE at the surface (Figure 4 h).

On the other hand, the responses of the middle- and low-level clouds are consistent with the SOR-forced equatorward shift of the eddy-driven westerly jet in the southern hemispheric mid-latitudes (see the review by Thompson et al. (2011)). This occurs especially during late spring and summer in the Southern Hemisphere. As the jet shifts, the associated storm track, precipitation, and cloud patterns follow. So cloud fraction decreases in the subtropical region (20°S-40°S), increases in the middle latitudes (40°S-60°S), and decreases in the polar region (higher than 60 degree). This then impacts the radiation budget, as documented by (Grise et al., 2013). As shown by the TOA radiative effect of cloud (Figure 3 d) and surface CRE (Figure 4 g), there are strong shortwave radiation anomalies that oscillate with the latitude. We find that these radiation anomalies are largely accounted for by the redistribution of liquid cloud and have very small southern-hemispheric mean values: 0.04 W m$^{-2}$ for surface radiation and -0.13 W m$^{-2}$ for TOA radiation.

Sea-ice response is important for the surface radiation budget in both polar regions. Arctic sea ice increases in boreal summer and autumn and Antarctic sea ice increases throughout the year (Figure 4 j). These increases cause considerable decreases in net shortwave radiation at surface, thus acting to cool surface temperature. Recent studies suggest that the Antarctic ozone hole has important influences on Antarctic sea ice (Sigmond and Fyfe, 2010;Bitz and Polvani, 2012;Smith et al., 2012). The large sea ice and radiation changes seen here affirm such ozone impact.

## 6 Discussion and conclusion

The Standard and NCNSI experiments conducted here suggest that clouds and sea ice are sensitive to stratospheric ozone perturbations and their radiative effects are critical for predicting surface temperature changes. Although the stratosphere-adjusted forcing of SOR is positive in both experiments, the warming effect of ozone recovery is offset by the cooling effect caused by high-cloud reduction and sea ice increase in the Standard experiment, which results in a weak global cooling. In addition, SOR also causes equatorward shift of jet stream, precipitation and mid- and low-clouds, especially in the southern hemisphere, which results in dipole patterns of zonal mean surface shortwave radiation anomalies and corresponding temperature anomalies.

The cloud and sea ice changes in the Standard experiment emerge as significant signals in response to the SOR forcing. The reduction of high clouds can be attributed to ozone-induced radiative warming and consequent relative humidity reduction in upper troposphere and lower stratosphere, in accordance with the findings of (Jenkins, 1999;Yang et al., 2012). The sea ice changes in the Arctic and around the Antarctic are influenced by ozone-induced indirect radiative effects, which are associated with the reduction of downward infrared radiation over the sea ice edge caused by the in-situ decreases of clouds and water vapor, and also the atmospheric cooling (Hu et al., 2016). The strong sea ice response to SOR forcing suggests the ongoing SOR would mitigate Antarctic sea-ice loss from greenhouse warming in 21$^{st}$ century (Smith et al., 2012).

Although an isolated SOR forcing as prescribed in our experiments is hypothetical, this forcing scenario makes a very unusual case of climate change in that the radiative forcing is positive (a warming effect) but the surface temperature response is



negative (cooling). The key factor that leads to the breakdown of the prediction appears to be a significant high cloud change

directly resulting from the forcing. Although this result is mainly based on one GCM, a suit of experiments and diagnoses here

suggest that this is a robust and significant rapid adjustment to SOR forcing and may have important implications such as for

geo-engineering. It warrants further research to examine whether such breakdown may occur to other types of forcing and/or

5   in reality.

**Acknowledgements**

We thank Timothy Merlis, Bruno Tremblay, and Jun Yang for their helpful comments and suggestions. Y. Xia and

Y. Huang are supported by grants from the Natural Sciences and Engineering Research Council of Canada (RGPIN 418305-

13) and the Fonds de recherche du Québec (NC-181248). Y. Hu is supported by the National Natural Science Foundation of

10   China (NSFC Grants No. 41530423 and 41375072).



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




**Table 1. Radiative forcing and adjustments, evaluated at the top of the atmosphere. The columns indicate the instantaneous radiative forcing (IRF) of O₃, the stratosphere-adjusted forcing (SAF), the effective radiative forcing (ERF, i.e., stratosphere and troposphere-adjusted forcing), the stratospheric adjustment, and the radiation changes caused by cloud, sea-ice, atmospheric temperature ($T_A$), water vapor (WV), and surface temperature ($T_S$), respectively. Unit: W m⁻².**

| | IRF of O3 | SAF | ERF | Stratospheric adjustment | Tropospheric/surface radiative effects | | | | |
| --- | --- | --- | --- | --- | --- | --- | --- | --- | --- |
| | | | | | Cloud | Sea-ice | $T_A$ | WV | $T_S$ |
| **NCNSI** | 0.49 | 0.30 | N/A | -0.19 | N/A | N/A | -0.77 | 0.77 | -0.26 |
| **Standard** | 0.60 | 0.29 | 0.01 | -0.31 | -0.39 | -0.10 | -0.04 | 0.10 | 0.08 |
| **Fixed-SST/SI** | 0.60 | 0.29 | 0.01 | -0.31 | -0.25 | N/A | -0.15 | 0.12 | N/A |





**Figures**

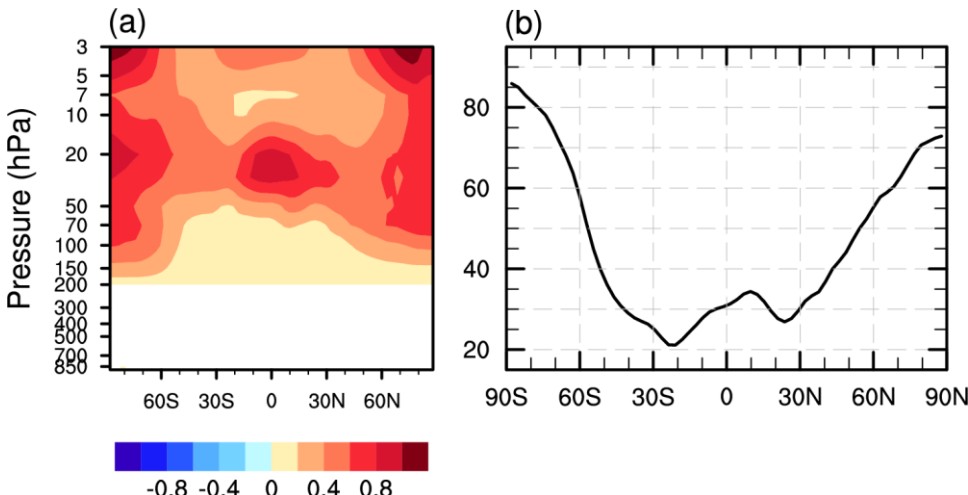

**Figure 1. The distribution of SOR. (a) The vertical cross section of the annual- and zonal-mean difference of ozone, unit: ppmv. (b)**

**The annual- and zonal-mean difference of total column ozone, unit: DU.**


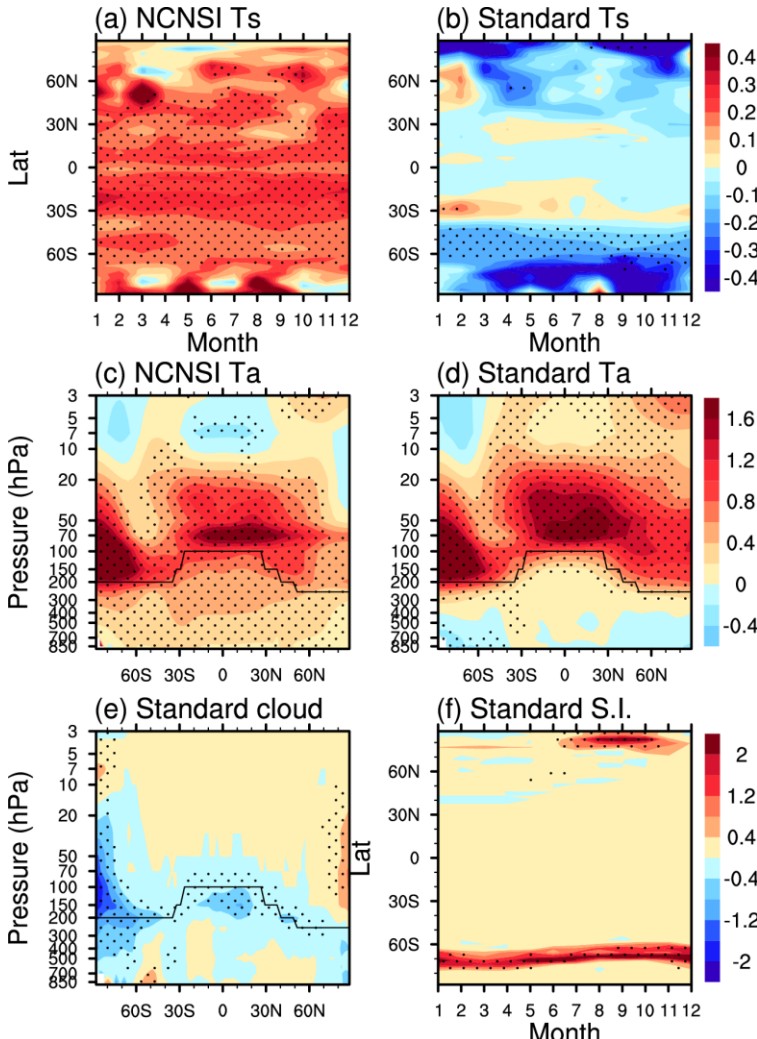

**Figure 2. Responses to SOR of zonal mean surface temperature, annual- and zonal-mean air temperature, annual- and zonal-mean cloud fraction, and zonal mean sea-ice fraction. Latitude-month distribution of surface temperature in the (a) NCNSI, and (b) Standard experiment. Vertical cross section of air temperature in the (c) NCNSI, and (d) Standard experiment. (e) Vertical cross section of cloud fraction, and (f) latitude-month distribution of sea-ice fraction in the Standard experiment. In (a, b), the color interval is 0.05 K. In (c, d), the color interval is 0.2 K. In (e-f), the color interval is 0.4%. Regions with dots are the places where differences have statistical significant levels higher than the 95 % confidence level (student t-test values are greater than 2.0). Black line in (c-e) indicates the tropopause of climatology.**



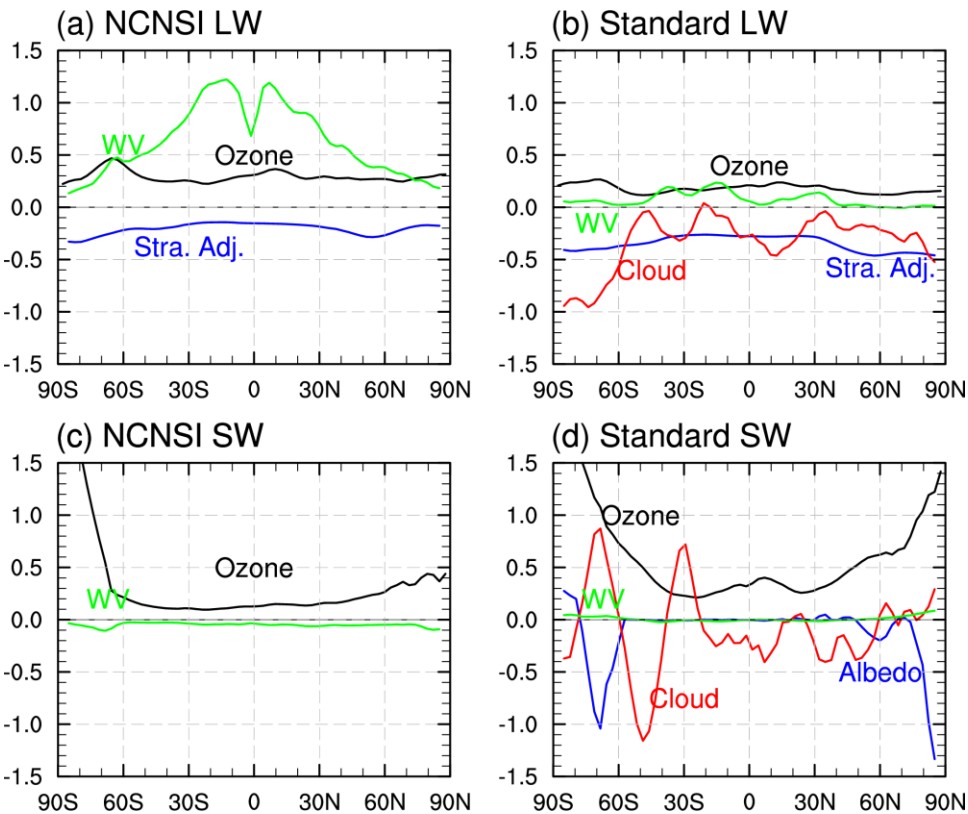

**Figure 3. Annual- and zonal-mean distribution of the radiative contributions at TOA for the NCNSI experiment: (a) longwave radiation: stratospheric temperature adjustment (blue line), ozone (black line), and water vapor (green line), (c) shortwave radiation: ozone (black line), and water vapor (green line). And for the Standard experiment: (b) longwave radiation: stratospheric adjustment (blue line), ozone forcing (black line), water vapor (green line), and radiative effect of cloud (red line); (d) shortwave radiation: ozone forcing (black line), water vapor (green line), cloud (red line) and ice-albedo (blue line) effects. Negative/positive values indicate upward/downward radiative flux at TOA. The radiative forcing of ozone are calculated with RRTMG.**







**Figure 4. The latitude-month distribution of the responses to SOR of the zonal mean surface radiation budget. (a) Net shortwave, (b) downward longwave, and (c) a+b in the NCNSI experiment. (d) Net shortwave, (e) downward longwave, and (f) d+e in the Standard experiment. (g) Shortwave CRE, (h) longwave CRE, and (i) g+h in the Standard experiment. (j) The albedo-induced surface radiation in the Standard experiment. Color interval is 0.5 W m$^{-2}$.**