# Peer review of "Strong modification of stratospheric ozone forcing by cloud and sea ice adjustments"

_Atmospheric Chemistry and Physics, 2016_

## Short Comment (SC1) · 23 Mar 2016

Thank you for this highly interesting study on the efficacy of ozone radiative forcings.

In reference to the comment in your short summary concerning previous studies on cloud adjustments to ozone forcing (also discussed in section 1 of your discussion paper), please see the opposing clear-sky and cloud radiative long-wave effects of upper tropospheric and lower stratospheric ozone changes and high clouds in

- Nowack, P. J., Abraham, N. L., Maycock, A. C., Braesicke, P., Gregory, J. M., Joshi, M. M., Osprey, A., and Pyle, J. A.: A large ozone-circulation feedback and its implications for global warming assessments, Nature Climate Change, 5, 41–45, doi:10.1038/nclimate2451, 2015.

[Figure]

Note in particular Figure 4 and the discussion on Supplementary Figure S6. Can you say more about the nature of the positive tropical ozone long-wave forcing you find?

You might further find the following studies interesting to compare to:

- Boer, G. J., Yu, B. Climate sensitivity and response, Climate Dynamics, 20, 415–429, doi: 10.1007/s00382-002-0283-3, 2003.

- Shindell, D., Faluvegi, G.: Climate response to regional radiative forcing during the twentieth century, Nature Geoscience, 2, 294–300, doi: 10.1038/ngeo473, 2009.

- Shindell, D. T., Faluvegi, G., Rotstayn, L., Milly, G.: Spatial patterns of radiative forcing and surface temperature response, Journal of Geophysical Research: Atmospheres, 120, 5385–5403, doi: 10.1002/2014JD022752, 2015. (and references therein)

- Stuber, N., Ponater, M., Sausen, R.: Why radiative forcing might fail as a predictor of climate change, Climate Dynamics, 24, 497–510, doi: 10.1007/s00382-004-0497-7, 2005.

Finally, you mention in section 2 the representation of coupling between wind stress and sea ice dynamics in the model. Do you know whether the choice of a slab ocean model as compared to a deep ocean model could affect dynamical atmosphere-ocean interactions? Out of interest, what part of the sea ice responses do you think is driven by the regional cloud forcings (in that sense the sea ice feedback and the regional cloud feedback are, as you say, partly related)?

---

## Referee Comment (RC1) · Anonymous Referee #1 · 8 Apr 2016

General Comments

Using experiments from a single global climate model (CAM3), this study examines the radiative effects of stratospheric ozone recovery. The key conclusion of the study is that, while the radiative forcing of stratospheric ozone recovery is positive (ozone recovery leads to a slight warming of the global-mean surface temperature), the radiative effects become negative when cloud and sea ice adjustments are included (ozone recovery leads to a slight cooling of the global-mean surface temperature). Overall, this is a useful contribution to the literature, and I recommend publication after the following revisions are addressed.

Major Revisions

This paper serves as a converse of the ozone depletion paper of Grise et al. (2013),

who concluded that, although stratospheric ozone depletion has a negative radiative forcing, the cloud changes due to stratospheric ozone depletion induce a net warming effect on the climate system. Given that both studies use the same CAM3 model and examine stratospheric ozone changes, it is surprising that the authors did not appreciate the strong connection between the two studies. This is especially important because a key shortcoming of the Grise et al. (2013) study was the use of CAM3. The clouds in the newest version of CAM (version 5) have substantially improved from version 3 (see Kay et al. 2012). As a result, subsequent studies have shown that the conclusions of Grise et al. (2013) are likely not robust, at least in terms of Southern Hemisphere mid-latitude clouds and how they respond to tropospheric eddy-driven jet shifts (Kay et al. 2014; Ceppi et al. 2014; Grise and Polvani 2014; Ceppi and Hartmann 2015). So, it is a bit perplexing that the authors of this study have chosen CAM3 for their analysis, as their cloud adjustment in this study is likely quite biased as a result.

It's probably beyond the scope of this paper to ask the authors to run additional simulations using different models, but perhaps the few historicalMisc runs from CMIP5 models that isolate stratospheric ozone depletion could provide some clues about inter-model spread (http://cmip-pcmdi.llnl.gov/cmip5/docs/historical_Misc_forcing.pdf). I would be highly surprised if the results from the CAM3 model are representative of all climate models (or the real world, for that matter). All that being said, this study is important because it shows that this effect occurs in at least some climate models, and the authors perform a much more rigorous diagnosis of the radiative effects of ozone recovery than in previous studies. I would just ask the authors to be very cautious about making any general conclusions about their results (as they do on the top of page 10), until a more comprehensive suite of models can verify them.

Specific (Minor) Revisions

Page 3, Lines 13-17: How does your methodology compare to the COOKIE experiments (http://www.euclipse.eu/downloads/Cookie.pdf ) used by previous studies? It sounds similar, but not exactly the same.

Page 3, Line 22: How realistic is the ERA-Interim ozone data compared to more commonly used satellite-derived ozone data sets? For reference, the ozone data used to force the CMIP5 models is provided at http://www.pa.op.dlr.de/CCMVal/AC&CSPARC_O3Database_CMIP5.html.

Page 6, Lines 8-10 (also Page 8, Lines 20-22): As stated above, it would useful to compare your numbers to the cloud-radiative effects for ozone depletion found by Grise et al. (2013) using the same model.

Page 8, Line 9: I don't understand the strong reduction in cloud cover in the Southern Hemisphere stratosphere in Fig. 2e. The absolute value of cloud cover and water vapor in the stratosphere should be very small here to begin with, so the changes seem too large to be physical. More explanation is warranted here. Perhaps this is also a deficiency of CAM3.

Page 9, Line 1: Why would Arctic sea ice increase a comparable amount as Antarctic sea ice, given that most of the ozone recovery should be in the Antarctic? Again, more explanation is warranted here.

Technical Corrections

Page 1, Line 17: Suggest changing "slow increasing" to "slowly increasing"

Page 2, Line 13: sophisticated GCMs

Page 6, Line 11: Reinstalled? Not sure what this means. Consider a different word choice.

Page 7, Line 19: Climatological

Figure 3 is barely discussed in the text. Is it essential to the paper? If so, it should be referenced and described in more detail.

References

[Figure]

Ceppi, P., and D. L. Hartmann, 2015: Connections between clouds, radiation, and midlatitude dynamics: A review. Curr. Clim. Change Rep., 1, 94–102.

Ceppi, P., M. D. Zelinka, and D. L. Hartmann, 2014: The response of the Southern Hemisphere eddy-driven jet to future changes in shortwave radiation in CMIP5. Geophys. Res. Lett., 41, 3244-3250, doi:10.1002/2014GL060043.

Grise, K. M., and L. M. Polvani, 2014: Southern Hemisphere cloud-dynamics biases in CMIP5 models and their implications for climate projections. J. Climate, 27, 6074–6092.

Kay, J. E., B. Medeiros, Y.-T. Hwang, A. Gettelman, J. Perket, and M. G. Flanner, 2014: Processes controlling Southern Ocean shortwave climate feedbacks in CESM. Geophys. Res. Lett., 41, 616-622, doi:10.1002/2013GL058315.

Kay, J., and Coauthors, 2012: Exposing global cloud biases in CAM using satellite observations and their corresponding instrument simulators. J. Climate, 25, 5190–5207.
* * *

---

## Short Comment (SC2) · 12 Apr 2016

[supplement omitted: unrelated document]

---

## Referee Comment (RC2) · Anonymous Referee #2 · 14 Apr 2016

Strong modification of stratospheric ozone forcing by cloud and sea ice adjustments

by Xia, Hu and Huang

The effect of stratospheric ozone recovery on simulated climate is examined using a global climate model (CAM3) coupled to a slab ocean model. In particular, the paper focuses on the result that the radiative forcing due to recovery is expected to warm the near-surface temperature but the simulated temperature change found to be reduced. This is attributed to the response of high clouds and sea-ice.

The paper presents a useful analysis of the response to changes in the stratospheric ozone in the simulations with this particular configuration. I suggest the paper be accepted with minor revisions. Below are comments and suggestions mainly related to the experiment design.

[Figure]

Major Comments

1. To examine the effect of sea-ice and clouds on the model response to stratospheric ozone recovery two simulations are performed, the control and one with no sea-ice and clouds that are invisible to radiation (NCNSI). While it may not substantially change the results of the analysis it seems that one would like to make incremental changes to isolate the effects of clouds and of sea-ice. For example, a set of with invisible clouds and a set with no sea-ice or perhaps with sea-ice invisible to radiation.

2. As noted in the Introduction work by McLandress et al, 2012 suggests that stratospheric ozone recovery my lead to surface cooling. Would it be possible to generalize and support the results found with CAM3 and comments made in the text by analyzing historical CMIP5 simulations that only vary ozone? For example, the list of models in Table 2 of Sigmond and Fyfe, 2013.

Sigmond, M. and Fyfe, J. C. The Antarctic Sea Ice Response to the Ozone Hole in Climate Models Journal of Climate, 2014, 27, 1336-1342

---

## Author Comment (AC1) · 6 May 2016

**Response to SC1**

We thank Mr. Nowack for his comments. Our responses are itemized below.

1. "In reference to the comment in your short summary concerning previous studies on cloud adjustments to ozone forcing (also discussed in section 1 of your discussion paper), please see the opposing clear-sky and cloud radiative long-wave effects of upper tropospheric and lower stratospheric ozone changes and high clouds in
   • Nowack, P. J., Abraham, N. L., Maycock, A. C., Braesicke, P., Gregory, J. M., Joshi, M. M., Osprey, A., and Pyle, J. A.: A large ozone-circulation feedback and its implications for global warming assessments, Nature Climate Change, 5, 41–45, doi:10.1038/nclimate2451, 2015.
   Note in particular Figure 4 and the discussion on Supplementary Figure S6. Can you say more about the nature of the positive tropical ozone long-wave forcing you find?"

**Response:** The ozone change in our paper is an idealized stratospheric ozone recovery (SOR) scenario (see Figure 1), in comparison to the ozone depletion in the upper troposphere and lower stratosphere in Nowack2015 (Figure 3a). The warming in the tropopause induced by SOR results in the decrease of high clouds in UTLS, which is consistent with the increase of cloud seen in Nowack2015 (Figure 4). We have pointed this out in Section 5 in the revised paper.

One aspect of our idealized ozone prescription is that the ozone change is positive throughout the stratosphere, including the tropical UTLS region. This renders very positive forcing across all the latitudes. We have also pointed this out in the revised paper.

2. Finally, you mention in section 2 the representation of coupling between wind stress and sea ice dynamics in the model. Do you know whether the choice of a slab ocean model as compared to a deep ocean model could affect dynamical atmosphere-ocean interactions?

**Response:** A number of previous works (DeConto et al., 2007;Cvijanovic and Caldeira, 2015) investigated the role of sea ice with the slab-ocean model. Particularly, Danabasoglu and Gent (2009) compared the slab ocean and the fully coupled configurations of CCSM3 (similar configuration to ours) and showed that the slab ocean setup provides a good estimate of the climate sensitivity of the fully coupled model. Moreover, we compared our CAM3-slab ocean simulations results to the coupled atmosphere-ocean simulations by CESM1 (CAM5), the climatology and variability of the sea ice extent have similar magnitude in the slab-ocean model. All these suggest the sea-ice responses simulated in our experiments are likely valid, although, as we have acknowledged in the revised

Conclusion Section, it warrants further research to test the robustness of the sea ice, as well as cloud, responses across different models and in reality.

3. Out of interest, what part of the sea ice responses do you think is driven by the regional cloud forcings (in that sense the sea ice feedback and the regional cloud feedback are, as you say, partly related)?

**Response:** The sea ice and clouds are coupled components in the high latitude climate system, which implies their feedbacks are potentially related. It is beyond the scope of this paper to elucidate how they are coupled. However, in an accompanying study of us (Hu et al., 2016), we find both cloud and sea ice responses to SOR tend to cool the local surface climate. We have mentioned this in the revised paper.

In our results, cloud-induced decrease of downward IR is only a small part of the total downward IR decrease, less than one-third.

Reference:

Cvijanovic, I., and Caldeira, K.: Atmospheric impacts of sea ice decline in CO2 induced global warming, Climate Dynamics, 44, 1173-1186, 10.1007/s00382-015-2489-1, 2015.

Danabasoglu, G., and Gent, P. R.: Equilibrium Climate Sensitivity: Is It Accurate to Use a Slab Ocean Model?, J Climate, 22, 2494-2499, 2009.

DeConto, R., Pollard, D., and Harwood, D.: Sea ice feedback and Cenozoic evolution of Antarctic climate and ice sheets, Paleoceanography, 22, n/a-n/a, 10.1029/2006PA001350, 2007.

Hu, Y., Xia, Y., Liu, J., and Huang, Y.: Stratospheric ozone-induced indirect radiative effects on Antarctic sea ice, To be submitted to Nature - Climate Change, 2016.

---

## Author Comment (AC2) · 6 May 2016

**Reply to RC1**

We thank the reviewer for his/her thoughtful and constructive comments that help improve the quality of our manuscript. We have incorporated the reviewer's suggestions in the revised manuscript. Our point-to-point response to the reviewer's comments are shown below.

5 Anonymous Referee #1:

**Major Revisions:**

1. This paper serves as a converse of the ozone depletion paper of Grise et al. (2013), who concluded that, although stratospheric ozone depletion has a negative radiative forcing, the cloud changes due to stratospheric ozone depletion induce a net warming effect on the climate system. Given that both studies use the same CAM3

10   model and examine stratospheric ozone changes, it is surprising that the authors did not appreciate the strong connection between the two studies.

**Response:** We thank the reviewer for pointing out the consistency between the results of our study and Grise et al. (2013), and have modified the paper to recognize the connections between the two studies. These include: line 12 in page 6, line 23 in page 8, and lines 3-4 in page 9.

15 2. So, it is a bit perplexing that the authors of this study have chosen CAM3 for their analysis, as their cloud adjustment in this study is likely quite biased as a result. It's probably beyond the scope of this paper to ask the authors to run additional simulations using different models, but perhaps the few historicalMisc runs from CMIP5 models that isolate stratospheric ozone depletion could provide some clues about inter-model spread (http://cmip-pcmdi.llnl.gov/cmip5/docs/historical_Misc_forcing.pdf). I would be highly surprised if the results

20   from the CAM3 model are representative of all climate models (or the real world, for that matter). All that being said, this study is important because it shows that this effect occurs in at least some climate models, and the authors perform a much more rigorous diagnosis of the radiative effects of ozone recovery than in previous studies. I would just ask the authors to be very cautious about making any general conclusions about their results (as they do on the top of page 10), until a more comprehensive suite of models can verify them.

25 **Response:** We thank the reviewer for cautioning us the potential deficiencies of the CAM3. The choice of CAM3 was because it had already been used in our previous research when this study began and also it takes less computing time to integrate compared to the later versions. We recognize the discrepancies especially concerning clouds in CAM3 compared to other models as pointed out by the reviewer.

Following this and the other reviewer's suggestion, we have analyzed the CMIP5 experiments. Five CMIP5 models, CCSM4, CESM1-CAM5, FGOALS-g2, GISS-E2-H, and GISS-E2-R, have ozone-only historical experiments, which, however, does not isolate the effects of stratospheric ozone depletion (http://cmip-pcmdi.llnl.gov/cmip5/docs/historical_Misc_forcing.pdf). We calculated, using RRTMG, the instantaneous forcing of ozone change from 1960 to 2000 to be negative: -0.20 W m$^{-2}$, although most models (except GISS-E2-R) show weak global warming (Figure R1). The global- and annual-mean sea ice and cloud changes are shown in Figures R2 and R3 respectively, both of which show statistically significant (stippled) responses, such as high level cloud increase and Antarctic sea ice reduction, to ozone forcing, although the pattern, magnitude and even sign of the changes are of noticeable inter-model differences, which supports the reviewer's point about inter-model spread. However, given that the forcing prescribed in the experiment is not exclusively stratospheric ozone change, these results may also reflect the complications of the impact of tropospheric ozone change.

In response to this important comment of the reviewer, we have acknowledged in the revised Conclusion Section that results presented here is based on only one model and it takes further research to verify its robustness. We also like to mention here that since the submission of this paper, we have started additional experiments, using different model configurations such as CESM1-CAM5 and different prescriptions of stratospheric ozone change. The preliminary results suggest that the high-cloud and sea ice responses as reported in this paper is at least qualitatively similar (robust) in these experiments. We intend to present these results in a following-up paper.

**Specific (Minor) Revisions:**

1. Page 3, Lines 13-17: How does your methodology compare to the COOKIE experiments (http://www.euclipse.eu/downloads/Cookie.pdf ) used by previous studies? It sounds similar, but not exactly the same.

**Response:** Our methodology is similar to the Clouds On Off Klima Intercomparison Experiment (COOKIE). We don't consider the cloud radiative effects, but consider cloud and precipitation in hydrological cycle including latent heat release, which is same as the COOKIE setup. We have noted the similarity to COOKIE in our experiment design in the revised paper.

2. Page 3, Line 22: How realistic is the ERA-Interim ozone data compared to more commonly used satellite-derived ozone data sets? For reference, the ozone data used to force the CMIP5 models is provided at http://www.pa.op.dlr.de/CCMVal/AC&CSPARC_O3Database_CMIP5.html.

**Response:** We have acknowledged in the revised paper that our ozone prescription represents an idealized (simplified) SOR scenario. One noticeable difference compared to the scenario used by CMIP5 is that the ozone

change is made positive (to increase) everywhere in the stratosphere, which renders nearly uniformly positive zonal mean forcing as shown in Figure 3 in the paper and simplifies the investigation. We have also acknowledged in the revised Conclusion Section that this is another aspect that warrants further investigation.

3. Page 6, Lines 8-10 (also Page 8, Lines 20-22): As stated above, it would useful to compare your numbers to the cloud-radiative effects for ozone depletion found by Grise et al. (2013) using the same model.

**Response:** The cloud-radiative effects for ozone depletion found by Grise et al. (2013) has been added in the revised manuscript, cf. line 12 in page 6.

4. Page 8, Line 9: I don't understand the strong reduction in cloud cover in the Southern Hemisphere stratosphere in Fig. 2e. The absolute value of cloud cover and water vapor in the stratosphere should be very small here to begin with, so the changes seem too large to be physical. More explanation is warranted here. Perhaps this is also a deficiency of CAM3.

**Response:** In theory, there can be many PSCs, at least seasonal ones, in the region under question, but we agree that, as the reviewer questions, the climatology as well as the response simulated by CAM3 may be too large. The mean cloud fraction can reach 20% in boreal autumn in the Antarctic lower stratosphere in CAM3; in comparison, it is about 10% in CCSM4 and 3% in CESM-CAM5. However, as there lacks strong observational constraints, it is difficult to rule out any of these simulations. As the region under question is small, this issue is unlikely to significantly affect the global mean forcing or warming/cooling values that we are concerned with in this paper, although we agree with the reviewer this is an aspect of the CAM3 simulation that needs to be further validated in future research.

5. Page 9, Line 1: Why would Arctic sea ice increase a comparable amount as Antarctic sea ice, given that most of the ozone recovery should be in the Antarctic? Again, more explanation is warranted here.

**Response:** Firstly, in our idealized ozone change scenario, the Arctic increase is comparable to the Antarctica. Secondly, we note that as evident from the analysis of CMIP5 models, there is much larger inter-model spread in terms of sea ice response to ozone forcing. We acknowledge this is an aspect that concerns the robustness of the response and is worth further investigation.

**Technical Corrections:**

1. Page 1, Line 17: Suggest changing "slow increasing" to "slowly increasing"

**Response:** It has been changed.

2. Page 2, Line 13: sophisticated GCMs

**Response:** It has been changed.

3. Page 6, Line 11: Reinstalled? Not sure what this means. Consider a different word choice.

5  **Response:** It has been changed to be "balanced".

4. Page 7, Line 19: Climatological

**Response:** It has been changed.

5. Figure 3 is barely discussed in the text. Is it essential to the paper? If so, it should be referenced and described in more detail.

10  **Response:** It has been referenced and described in more details in the revised manuscript, cf. line 13 and line 21 in page 5.

Reference:

Bitz, C. M., and Polvani, L. M.: Antarctic climate response to stratospheric ozone depletion in a fine resolution ocean climate model, Geophysical Research Letters, 39, 2012.

Grise, K. M., Polvani, L. M., Tselioudis, G., Wu, Y., and Zelinka, M. D.: The ozone hole indirect effect: Cloud-radiative anomalies accompanying the poleward shift of the eddy-driven jet in the Southern Hemisphere, Geophys Res Lett, 40, 3688-3692, 10.1002/grl.50675, 2013.

Haumann, F. A., Notz, D., and Schmidt, H.: Anthropogenic influence on recent circulation-driven Antarctic sea ice changes, Geophysical Research Letters, 41, 8429-8437, 2014.

Neale, R. B., Chen, C.-C., Gettelman, A., Lauritzen, P. H., Park, S., Williamson, D. L., Conley, A. J., Garcia, R., Kinnison, D., and Lamarque, J.-F.: Description of the NCAR community atmosphere model (CAM 5.0), NCAR Tech. Note NCAR/TN-486+ STR, 2010.

Polvani, L. M., and Smith, K. L.: Can natural variability explain observed Antarctic sea ice trends? New modeling evidence from CMIP5, Geophysical Research Letters, 40, 3195-3199, 2013.

Sigmond, M., and Fyfe, J. C.: Has the ozone hole contributed to increased Antarctic sea ice extent?, Geophysical Research Letters, 37, 2010.

Sigmond, M., and Fyfe, J. C.: The Antarctic Sea Ice Response to the Ozone Hole in Climate Models, Journal of Climate, 27, 1336-1342, 2014.

Smith, K. L., Polvani, L. M., and Marsh, D. R.: Mitigation of 21st century Antarctic sea ice loss by stratospheric ozone recovery, Geophysical Research Letters, 39, 2012.

Turner, J., Bracegirdle, T. J., Phillips, T., Marshall, G. J., and Hosking, J. S.: An Initial Assessment of Antarctic Sea Ice Extent in the CMIP5 Models, Journal of Climate, 26, 1473-1484, 2013.

Figures:

[Figure]

Figure R1. Zonal-mean surface temperature trends from 1960 to 2000 for historicMisc ozone only runs from (a)
CCSM4, (b) CESM1-CAM5, (c) FGOALS-g2, (d) GISS-E2-H, and (e) GISS-E2-R, unit: K/40 yrs.

[Figure]

Figure R2. Zonal-mean trends of sea ice fraction from 1960 to 2000 for historicMisc ozone only runs from (a) CCSM4, (b) CESM1-CAM5, (c) FGOALS-g2, (d) GISS-E2-H, and (e) GISS-E2-R, unit: %.

[Figure]

Figure R3. Zonal- and annual-mean cloud fraction trends from 1960 to 2000 for historicMisc ozone only runs from (a) CCSM4, (b) CESM1-CAM5, (c) FGOALS-g2, (d) GISS-E2-H, and (e) GISS-E2-R, unit: %.

---

## Author Comment (AC3) · 6 May 2016

**Response to SC2**

We thank A. F. Tuck for his comments. Our response are itemized here.

1. The authors might like to consider the conclusions reached in the attached .pdf, which examined factors like cloud cover, surface nature and temperature, and the important influence of actual local observations of ozone and water vapour. The use of matching observed outgoing long wave radiation to underlying cloud was a useful innovation.

**Response:** We thank Dr. Tuck for his comments. We recognize the importance of PSC and its interactions with ozone, as well as other atmospheric and surface variables, which jointly defines polar climate and potentially affects climate of greater region through modifying heating rate and atmospheric temperature structure, as discussed by Hicke and Tuck (2001). Such interactions are accounted for in our simulations to the extent that PSCs are simulated in the CAM3.

Reference:

Hicke, J., and Tuck, A.: Polar stratospheric cloud impacts on Antarctic stratospheric heating rates, Q J Roy Meteor Soc, 127, 1645-1658, 10.1002/qj.49712757510, 2001.

---

## Author Comment (AC4) · 6 May 2016

**Reply to RC2**

We thank the reviewer for his/her thoughtful and constructive comments that help improve the quality of our manuscript. We have incorporated the reviewer's suggestions in the revised manuscript. Our point-to-point response to the reviewer's comments are shown below.

5

**Anonymous Referee #2:**

**Major Comments:**

- 1. To examine the effect of sea-ice and clouds on the model response to stratospheric ozone recovery two simulations are performed, the control and one with no sea-ice and clouds that are invisible to radiation
- 10 (NCNSI). While it may not substantially change the results of the analysis it seems that one would like to make incremental changes to isolate the effects of clouds and of sea-ice. For example, a set of with invisible clouds and a set with no sea-ice or perhaps with sea-ice invisible to radiation.

**Response:** We thank the reviewer for this good suggestion.

Following the reviewer's suggestion, to isolate the effects of clouds and of sea-ice, two sets of experiments are conducted. In the first set, we set all the cloud fractions to zero in radiative heating rate and flux calculations and thus suppress the radiative effects of clouds. In the other set of integrations, we set the freezing temperature to 180 degree centigrade so that there is effectively no sea ice in the simulation. These two sets of experiments are denoted as "No Cloud (NC)" and "No Sea-Ice (NSI)" respectively in the following.

As in the other experiments documented in the paper, in order to examine the impact of SOR on surface temperature,
two 100-year integrations, prescribed with identical concentrations of well-mixed greenhouse gases (CO2, CH4, N2O, etc.) but different stratospheric ozone concentrations, are conducted in both the NC and the NSI experiments. We assess the SOR impacts by contrasting the means of appropriate variables in the last 85 years of two 100-year simulations (the difference between two equilibrium states).

As shown by Figure R1, the surface temperature response to SOR is 0.18 K in the NC experiment (Figure R1 a), which is similar to that in the NCNSI experiment. The SOR warms not only the stratosphere but also the troposphere (Figure R1 b). We also find non-significant sea ice depletion in both hemispheres, which is opposite to that in the Standard experiment (Figure R1 c). So if there were no clouds, the sea ice only makes a small effect on the stratospheric ozone forcing.

The NSI experiment confirmed the above finding. If there were no sea ice but still clouds, a near-zero global surface warming (about 0.03 K) is resulted (Figure R2 a). The warming in the troposphere in the NSI experiment is also reduced compared to the NC and NCNSI experiments (Figure R2 b). We can also see the reduction of high clouds in the UTLS, which is consistent with that in the Standard experiment (Figure R2 c).

- 5 Hence, in summary, as we have concluded, the clouds have more important impacts on the modification of stratospheric ozone forcing than the sea ice. We have added the new experiment results in the relevant texts (lines 10-14 in page 9).
- As noted in the Introduction work by McLandress et al, 2012 suggests that stratospheric ozone recovery my lead to surface cooling. Would it be possible to generalize and support the results found with CAM3 and comments made in the text by analyzing historical CMIP5 simulations that only vary ozone? For example, the list of models in Table 2 of Sigmond and Fyfe, 2013.

**Response:** We thank the reviewer for this suggestion. Following the suggestion, we have analyzed the CMIP5 experiments. Five CMIP5 models, CCSM4, CESM1-CAM5, FGOALS-g2, GISS-E2-H, and GISS-E2-R, have

- 15 ozone-only historical experiments, which, however, does not isolate the effects of stratospheric ozone depletion (http://cmip-pcmdi.llnl.gov/cmip5/docs/historical\_Misc\_forcing.pdf). We calculated, using RRTMG, the instantaneous forcing of ozone change from 1960 to 2000 to be negative: -0.20 W m-2, although most models (except GISS-E2-R) show weak global warming (Figure R3). The global- and annual-mean sea ice and cloud changes are shown in Figures R4 and R5 respectively, both of which show statistically significant (stippled)
- 20 responses, such as high level cloud increase and Antarctic sea ice reduction, to ozone forcing, although the pattern, magnitude and even sign of the changes are of noticeable inter-model differeces. Given that the forcing presceribed in the experiment is not exclusively stratospheric ozone change, these results do not lead to conclusive assessment. We have acknowledged in the revised manuscript that it takes further research to elucidate whether and how SOR leads to global warming or cooling in reality.

25

Figures:

Figure R1. Responses to SOR of (a) zonal mean surface temperature, (b) annual- and zonal-mean air temperature, (c) zonal mean sea-ice fraction in the NC experiment.

5

---

## Author Response (AR2)

**Reply to RC1**

We thank the reviewer for his/her further comments. We have incorporated the reviewer's suggestions in the revised manuscript. Our responses to the reviewer's comments are summarized below.

Anonymous Referee #1:

5   **Minor Revisions:**

1.  I still feel that the revised manuscript does not adequately acknowledge the limitations of the clouds in the CAM3 model. As I stated in my previous comments, readers need to be aware that the clouds in this model have well known deficiencies, which have been well documented by previous studies (see detailed discussion and citations in my earlier comments). The authors address limitations of the sea ice in the model in section 2,

10     so the same should be done for the clouds. Additionally, in response to the reviewer comments, the authors have done some nice analyses with the CMIP5 ozone-only historical runs, and while I don't think those figures need to go into this paper, the results should be described briefly in the last paragraph of the paper when the authors are discussing the robustness of their results.

**Response:** Following the reviewer's suggestions, we have added the limitations of the clouds in the CAM3 model

15  in Section 2 (lines 7-14 on page 3) and expanded the relevant discussion in the Discussion and Conclusion Section (lines 22-23, on page 10). We have also added the description of the results of the CMIP5 ozone-only historical runs in the Discussion and Conclusion (lines 1-5 in page 11).

**Strong modification of stratospheric ozone forcing by cloud and sea ice adjustments**

Y. Xia[1], Y. Hu[2], and Y. Huang[1]

[1]Department of Atmospheric and Oceanic Sciences, McGill University, Montreal, Canada

5  [2]Department of Atmospheric and Oceanic Sciences, Peking University, Beijing, China

*Correspondence to*: Y. Xia (yan.xia3@mail.mcgill.ca)

**Abstract.**  We investigate the climatic impact of stratospheric ozone recovery (SOR) with a focus on the surface temperature change in atmosphere-slab-ocean coupled climate simulations. We find that although SOR would cause significant surface warming (global mean: 0.2 K) in a climate free of clouds and sea-ice, it causes surface cooling (-0.06 K) in the real climate.

10  The results here are especially interesting in that the stratosphere-adjusted radiative forcing is positive in both cases. Radiation diagnosis shows that the surface cooling is mainly due to a strong radiative effect resulting from significant reduction of global high clouds and, to a lesser extent, from an increase in high-latitude sea ice. Our simulation experiments suggest clouds and sea ice are sensitive to stratospheric ozone perturbation, which constitutes a significant radiative adjustment that influences the sign and magnitude of the global surface temperature change.

15  **1 Introduction**

Observational records show that stratospheric ozone has declined prior to the late 1990s and then started stabilizing and even slowly increasing, especially in the Polar Regions (WMO, 2007, 2011). It is expected that the ozone layer would return to the pre-1980 level in the 2050s (Bekki, 2011). It is known that ozone is a greenhouse gas, and that stratospheric ozone has a warming effect on tropospheric-surface climate, which has been demonstrated by early simulation works with radiative-

20  convective models (Ramanathan and Dickinson, 1979;Lacis et al., 1990). Consistent with such understanding, ozone depletion generally leads to a negative radiative forcing (after accounting for stratospheric temperature adjustment) that cools the climate (Forster and Shine, 1997;Hansen et al., 2005;Conley et al., 2013;Myhre et al., 2013;Macintosh et al., 2016). On such basis,

one would expect that stratospheric ozone recovery (SOR) exerts a positive forcing that should lead to troposphere and surface warming. The single-column simulation by Hu et al. (2011) agrees with such expectation, although their efforts to distinguish the responses to SOR in full general circulation models (GCMs) is impeded by climate sensitivity differences between the two groups of models (McLandress et al., 2012). Very interestingly, McLandress et al. (2012) show a weak troposphere-surface cooling in response to SOR in a coupled chemistry-climate model (CCM). As presented below, such a weak cooling is also seen in our simulation with an atmospheric GCM coupled to a slab-ocean model. These results raise important questions: how does surface cooling result from the positive radiative forcing of SOR in GCM simulations? Why do GCMs and radiative-convection models yield opposite results? In this paper, we are motivated to answer these questions and reconcile the contradiction of the warming prediction based on single-column model simulations.

One prominent deficiency of the one-dimensional radiative-convective models is that they neglect effects of clouds as well as snow and ice albedo. Thus, results from these simplified models may not realistically represent the responses to SOR. Hence, our hypothesis is that the radiative adjustment of clouds and sea ice may override the forcing of SOR and change the direction of surface temperature change in more sophisticated GCMs. To test this hypothesis, we perform two sets of SOR forcing experiments using a three-dimensional climate model, one with standard settings and the other with cloud and sea-ice artificially removed in the simulation. Comparison of the two sets of simulations shall elucidate the effects of cloud and sea ice. In the following sections, we will describe the configuration and results of these experiments, dissect the simulations from a radiative budget perspective, and summarize our main findings in order.

**2 Model and experiment design**

Here, we conduct and analyze a series of SOR experiments using the NCAR Community Atmosphere Model, version 3 (CAM3) coupled with a Slab Ocean Model (SOM) (Collins et al., 2006;Neale et al., 2010). All of the runs presented below are made with T42 horizontal resolution (~2.8°x2.8°) and coupled to a 50-meter-deep SOM. The SOM configuration uses a simple ocean component (Kiehl et al., 2006;Danabasoglu and Gent, 2009), combined with a thermodynamic sea ice component that is based on the Community Sea Ice Model (CSIM5, (Briegleb, 2004)) and allows for a fully-interactive treatment of surface exchange

processes in CAM3. Danabasoglu and Gent (2009) compare the slab ocean and the fully coupled configurations of CCSM3 and find that the slab ocean setup provides a good estimate of the climate sensitivity of the fully coupled model. Although the slab-ocean component lacks explicit representation of ocean currents, GCM surface winds drive the sea ice dynamics, with advection simulated as a cavitating fluid (Flato and Hibler, 1990, 1992). Compared with the coupled atmosphere-ocean simulations by CESM1 (CAM5), the annual cycle of climatological sea ice extent has similar magnitude (varying from 3 to $15 \times 10^6$ km$^2$) in SOM. The variabilities of the annual-mean sea ice extent are also similar (about $2\text{-}3 \times 10^6$ km$^2$) in SOM and coupled atmosphere-ocean simulations. A few studies have documented the deficiencies in the CAM3-simulated cloud fields, including the biases in climatological mean cloud fraction and optical depth and cloud responses to tropospheric eddy-driven jet shifts, especially in the Southern Hemispheric mid-latitude region (Kay et al., 2012;Ceppi et al., 2014;Grise and Polvani, 2014;Kay et al., 2014;Ceppi and Hartmann, 2015). We also notice that the climatological mean fraction of the polar stratospheric clouds in the Antarctica is noticeably larger in the CAM3 (~15% in austral spring) than in a few other models (CCSM4: ~10% and CESM-CAM5: ~3%). However, as the region under question is small, it is unlikely to significantly affect the global mean cloud radiative effect that we are concerned with in this paper. This will be further discussed in the concluding section.

[revised manuscript text omitted]

2012;Nowack et al., 2015). An examination of the CMIP5 ozone-only historical hindcast experiments shows that there are noticeable high-cloud changes in many models and the majority of the models show a weak warming in response to the significant ozone depletion from 1960 to 2000, which is in accordance with the results reported here; however, because the forcing prescribed in the experiments is not exclusively stratospheric ozone change, these results do not lead to conclusive assessment. It warrants further research to verify whether the cloud and sea ice responses to stratospheric ozone are robust across different GCMs and whether the responses are sensitive to details in the prescription of ozone change.

**Acknowledgements**

We thank Timothy Merlis, Bruno Tremblay, and Jun Yang for their helpful comments and suggestions. Y. Xia and Y. Huang acknowledge the grants from the Natural Sciences and Engineering Research Council of Canada (RGPIN 418305-13) and the Fonds de recherche du Québec (NC-181248). Y. Xia also acknowledges a fellowship from China Scholarship Council (CSC, No. 201405990230). Y. Hu acknowledges the grants from the National Natural Science Foundation of China (NSFC Grants No. 41530423 and 41375072).

已移动(插入)〔1〕

已上移 [1]: from China Scholarship Council (CSC, No. 201405990230).

[revised manuscript text omitted]